# Flexural Strength of Resin Core Build-Up Materials: Correlation to Root Dentin Shear Bond Strength and Pull-Out Force

**DOI:** 10.3390/polym12122947

**Published:** 2020-12-09

**Authors:** Masao Irie, Yukinori Maruo, Goro Nishigawa, Kumiko Yoshihara, Takuya Matsumoto

**Affiliations:** 1Department of Biomaterials, Okayama University Graduate School of Medicine, Dentistry and Pharmaceutical Science, 2-5-1, Shikata-cho, Kita-ku, Okayama 700-8525, Japan; tmatsu@md.okayama-u.ac.jp; 2Department of Occlusion and Removable Prosthodontics, Okayama University, 2-5-1, Shikata-cho, Kita-ku, Okayama 700-8558, Japan; ykmar@md.okayama-u.ac.jp (Y.M.); goro@md.okayama-u.ac.jp (G.N.); 3National Institute of Advanced Industrial Science and Technology (AIST), Health and Medical Research Institute2217-14 Hayashi-cho, Takamatsu, Kagawa 761-0395, Japan; kumiko.yoshihara@aist.go.jp

**Keywords:** flexural strength, resin core build-up materials, durability, pull-out force, bond strength

## Abstract

The aims of this study were to investigate the effects of root dentin shear bond strength and pull-out force of resin core build-up materials on flexural strength immediately after setting, after one-day water storage, and after 20,000 thermocycles. Eight core build-up and three luting materials were investigated, using 10 specimens (*n* = 10) per subgroup. At three time periods—immediately after setting, after one-day water storage, and after 20,000 thermocycles, shear bond strengths to root dentin and pull-out forces were measured. Flexural strengths were measured using a 3-point bending test. For all core build-up and luting materials, the mean data of flexural strength, shear bond strength and pull-out force were the lowest immediately after setting. After one-day storage, almost all the materials yielded their highest results. A weak, but statistically significant, correlation was found between flexural strength and shear bond strength (r = 0.508, *p* = 0.0026, *n* = 33). As the pull-out force increased, the flexural strength of core build-up materials also increased (r = 0.398, *p* = 0.0218, *n* = 33). Multiple linear regression analyses were conducted using these three independent factors of flexural strength, pull-out force and root dentin shear bond strength, which showed this relationship: Flexural strength = 3.264 × Shear bond strength + 1.533 × Pull out force + 10.870, *p* = 0.002). For all the 11 core build-up and luting materials investigated immediately after setting, after one-day storage and after 20,000 thermocycles, their shear bond strengths to root dentin and pull-out forces were correlated to the flexural strength in core build-up materials. It was concluded that the flexural strength results of the core build-up material be used in research and quality control for the predictor of the shear bond strength to the root dentin and the retentive force of the post.

## 1. Introduction

An endodontically-treated tooth presents a higher risk of biomechanical failure than a vital tooth. An appropriate restoration for endodontically–treated teeth is guided by both strength and esthetics. Posts are generally used to restore missing tooth structure and pulpless teeth [1], and new tooth-colored posts have improved the esthetics of teeth restored with posts and cores [2,3]. To ensure the durability of endodontically–treated teeth, it is extremely important that posts are optimally bonded to reduce debonding, long-term sorption and solubility, and fracture risks [4,5,6].

A core build-up system usually comprises a post, and it restores the tooth to the extent necessary to support a crown or an abutment tooth. Therefore, a core build-up material, such as resin composite, is a restoration placed in a badly broken-down tooth to restore the bulk of the tooth’s coronal portion. Various types of bonding systems have been used with different luting cements and core build-up materials [7]. Improvements in resin composites and advances in tooth substrate bonding systems have enabled the employment of more conservative techniques, which seek to maximally preserve the vitality of badly broken-down permanent premolar or molar teeth in their restoration. To provide post retention and improve the overall resistance of the root against fracture, resin composite core build-up materials are now widely used with an adhesive system [7,8,9,10,11,12,13,14,15,16].

Recently, self-adhesive resin cements which do not require any kind of tooth surface treatment have been developed. Due to their ease of use and low technique sensitivity, they are especially well suited for fiber post bonding in the root canals. Fiber posts are deep and narrow, thus making them a formidable challenge for any adhesive application protocol [17,18,19].

To increase bond strength at the post-core interface, both chemical and micromechanical surface pretreatments of fiber posts have been proposed. Chemical post surface pretreatments that are clinically employed today involve coating the post with a silane primer and/or a bonding agent. The most common silane coupling agent used in dentistry is a gamma-MPTS (gamma-methacryloxy propyl trimethoxy silane) that is diluted in an ethanol–water solution. Its working mechanism is to increase surface wettability, resulting in chemical bridge formation between the glass phase of the post and the resin matrix of the adhesive resin or luting cement [13,16,20,21,22,23].

The mechanical properties of restorative filling materials and luting agents have been evaluated using in vitro flexural testing [24,25,26,27,28,29,30,31,32,33]. In our previous studies [27,32], dental restorations which were polished after one-day storage following light activation showed improved shear bond strength and flexural properties, and thus improved marginal integrity. For luting agents, their shear bond strengths to dentin and flexural moduli increased after one-day storage, coupled with markedly decreased incidence of interfacial gaps [30]. Although more resin core build-up materials have emerged in the market, the relationship between flexural strength and shear bond strength to root dentin or pull-out force remains to be well understood. The aim was to investigate correlation between shear bond strength, pull-out force of resin core build-up to root dentin and flexural strength. Therefore, this investigation was carried out with multiple types of dual-cured luting agents and core build-up materials for 3 periods (immediately after curing, after 1-day storage, and after 20,000 thermocycles) (abbreviated as TC 20k)) [32,33] to assess their performance in terms of: (a) Flexural strength; (b) shear bond strength to root dentin; and (c) push-out force between core build-up material and fiber post. The hypothesis to be tested was that among the set of materials investigated, one or both of properties (b) and (c) would correlate to property (a).

## 2. Materials and Methods

The manufacturers and compositional details of eight core build-up materials and three luting materials (to be used as a core build-up material as recommended by their manufacturers), together with their pretreatment agents and fiber posts, are summarized in Table 1, Table 2 and Table 3. All procedures were performed in accordance with the manufacturers’ instructions. For light activation, a light curing unit (New Light VL-II, GC, Tokyo, Japan; optic diameter: 8 mm) was used. Before each application to the materials, light irradiance was checked using a radiometer (Demetron Kerr, Danbury, CT, USA). During the experiment, light irradiance was maintained at 450 mW/cm^2^.

Human premolars extracted for orthodontic reasons were used. After extraction, the teeth were immediately stored in cold, distilled water at about 4 °C for 1 to 2 months until use. The research protocol of this study was approved by the Ethics Committee of Okayama University Graduate School of Medicine, Dentistry and Pharmaceuticals Sciences, and Okayama University Hospital (No. 1508-007). For premolars extracted used is the study; was there any exclusion criteria, such as teeth with a caries facture line or worn, were excluded.

A single operator performed all the bonding procedures as described in Table 2. Ten specimens were made for each material, time period, and the mechanical property investigated. All procedures, except for cavity preparation and mechanical testing, were performed in a thermo-hygrostatic room kept at 23 ± 0.5 °C and 50 ± 2% relative humidity.

### 2.1. Flexural Strength

Teflon molds (25 × 2 × 2 mm) were used to prepare the resin core build-up specimens (*n* = 10/group). Each specimen was cured in three overlapping sections, with each section cured for 20 s. Flexural strength was measured at three different time periods: Immediately after light activation (or setting), after one-day distilled water storage at 37 °C, and after 20,000 thermocycles (thermal stress between 5 and 55 °C; 1 min dwell time; abbreviated as TC 20k) [31,32]. Data were acquired using a 3-point bending test with a 20-mm span and a load speed of 0.5 mm/min (5565, Instron, Canton, MA, USA) as outlined in ISO 9917-2 (1996), and then the measurements were calculated (Series IX software, Instron, Canton, MA, USA) [32].

### 2.2. Shear Bond Strength to Root Dentin

Shear bond strengths of resin core build-up materials (*n* = 10/group) to flat root dentin surfaces were determined at three time periods: Immediately after light activation, after one-day distilled water storage at 37 °C, and after 20,000 thermocycles (thermal stress between 5 and 55 °C; 1 min dwell time; abbreviated as TC 20k). Specimens (*n* = 10/group) were obtained from human premolars and molars embedded in slow-setting epoxy resin (EpoFix Resin, Struers, Copenhagen, Denmark). Flat root dentin surfaces were obtained by grinding with wet silicon carbide paper (#600), then pretreated with a conditioner/primer as recommended by the manufacturer or with a silanized primer as described in Table 2. Each resin core build-up material was placed into a Teflon mold (diameter: 3.6 mm, height: 2.0 mm) set on a root dentin surface, and light-activated as described above. The specimens thus obtained were mounted on a universal testing machine (5565, Instron, Canton, MA, USA), and shear stress was applied at a crosshead speed of 0.5 mm/min. After the shear bond strength measurements, all failed specimens were analyzed using a light microscope (4×; SMZ-10, Nikon, Tokyo, Japan) to determine the nature of their fractures [9,32].

### 2.3. Pull-Out Force

Each post was pretreated with a conditioner/primer according to the manufacturer’s instructions or with a silanized primer (Table 2). With Kerr’s system, BeautiCore FiberPost (diameter: 1.6 mm; Shofu, Kyoto, Japan; Table 2) was used as it had no recommended post system. Each resin core build-up material was placed into a Teflon mold (upper diameter: 8 mm, bottom diameter: 3.6 mm, height: 5 mm; Figure 1) set on a glass plate precoated with Vaseline. Then, each post was inserted at the center of the Teflon mold using a retainer and cured in four overlapping sections, with each section cured for 20 s as described above.

Specimens thus obtained were mounted on a universal testing machine (5565, Instron, Canton, MA, USA), and pull-out force was applied at a crosshead speed of 0.5 mm/min (*n* = 10/group; Figure 1). As each post differed in its external form, the maximum failure load was recorded in newton (N). After the pull-out force measurements, all failed specimens were analyzed using a light microscope (200×; Measurescope MM-II, Nikon, Tokyo, Japan) to determine the nature of their fractures. Three categories of failure mode were evaluated: (1) Adhesive failure at the interface between post and resin core material; (2) cohesive failure within the resin core material; and (3) combination of adhesive and cohesive failures on the same surface or a mixed failure.

### 2.4. Statistical Analysis

Statistical analysis was performed using the software package, Statistica 9.1 (Statsoft, OK, USA).

Analysis of variance (two-way ANOVA) with Tukey-HSD for post-hoc comparison was used to analyze the data obtained for flexural strength, shear bond strength, and pull-out force (*p* < 0.05). For shear bond strength to root dentin and pull-out force analyses, two-way ANOVA with Tukey-HSD for post-hoc comparison was used to analyze the significant differences (*p* < 0.05).

Any possible correlations among flexural strength, shear bond strength to root dentin, and pull-out force were also evaluated (Spearman; *p* < 0.05). Analyses were conducted using SPSS version 19 (Chicago, IL, USA). Multiple linear regression analyses were conducted using the three independent factors of flexural strength, pull-out force, and root dentin shear bond strength. In addition, partial correlation was evaluated for among flexural strength, shear bond strength to root dentin and pull-out force.

## 3. Results

### 3.1. Flexural Strength

Flexural strength data and their statistical analysis results are given in Table 4, Table 5 and Table 6. Except RelyX Unicem 2 Automix, the flexural strength of all core build-up materials changed significantly with time (*p* < 0.05). The one-day time period yielded the highest mean data, except for Clearfil DC Core Automix ONE. For all the three time periods, ESTECORE showed the highest values among all the core build-up materials.

### 3.2. Shear Bond Strength to Root Dentin

Shear bond strength data and their statistical analysis results are given in Table 7, Table 8 and Table 9. The shear bond strength of many core build-up materials did not change significantly with time (*p* > 0.05). The one-day time period yielded the highest mean data, except for MultiCore Flow. For all the three time periods, RelyX Unicem 2 Automix showed the lowest values. In contrast, MultiCore Flow showed the highest values at both immediately-after-setting and after TC 20k time periods.

On failure mode, no adhesive fractures were observed. In general, the proportion of adhesive failure modes was the same in three conditions.

### 3.3. Pull-Out Force

Pull-out force data and their statistical analysis results are given in Table 10, Table 11 and Table 12. The pull-out force of many core build-up-post systems did not change significantly with time (*p* > 0.05). The one-day time period yielded the highest mean data. ESTECORE system showed the highest values at both the immediately-after-setting and after TC 20k time periods, while MultiCore Flow system was the lowest at these time periods.

On failure mode, a few groups showed adhesive fracture. In general, the proportion of adhesive failures was similar at all the three time periods.

### 3.4. Correlation

For all materials at the three time periods (N = 33), relationships between the flexural strength data and either shear bond strength to root dentin or pull-out force were analyzed. These relationships were reproduced as graphs (Figure 2 and Figure 3). Flexural strength had a weak correlation with shear bond strength (r = 0.508, *p* = 0.0026; Figure 2); a weak positive correlation was also obtained with pull-out force (r = 0.398, *p* = 0.0218; Figure 3). Multiple linear regression analyses were conducted using the three independent factors of flexural strength, pull-out force and root dentin shear bond strength, which showed this relationship: Flexural strength = 3.264 × Shear bond strength + 1.533 × Pull out bond strength + 10.870, *p* = 0.002 (Figure 4). No positive correlation was obtained between shear bond strength and pull-out force (*p* = 0.102). Regardless of whether the build-up material was made of resin core or luting cement, flexural strength was correlated to root dentin shear bond strength and pull-out force.

## 4. Discussion

The core build-up system is a harmony of root dentin surface, core materials and the surface of post. This situation is the results of interactions of a layer of intermediate materials with two surfaces producing two adhesive interfaces. In the many cases, the contact area between the core build-up materials involved may be much greater because of a mechanically rough interface. The type of shear bond strength testing is categorized in terms of the mechanical loading direction. Almost shear bond strength testing are categorized as shear or tensile bond strength. For shear bond strength testing, flexural strength is very important [8,24,25,26,27]. Flexural strength testing is sensitive to surface defects such as cracks, voids, and scratches, which can influence the fracture characteristics of a brittle material [15]. The degree of high flexural strength is believed to reflect high resistance to surface defects and erosion. Therefore, it is thought that flexural strength is a significant important mechanical property of resin core build-up and luting materials. In the bid to better these materials, ensuing research and development efforts should focus on the change of flexural strength with elapsed time [29,30,31,32,33,34]. Therefore, this investigation was carried out with multiple types of core build-up and luting materials at three different time periods to evaluate their flexural strength performance in relation to their shear bond strength to root dentin and the push-out force between post and core.

The first test hypothesis was accepted in that bond strength to root dentin was found to correlate to flexural strength (r = 0.508, *p* = 0.0026). Most of the fracture modes seen after shear bond testing were mixed and cohesive failures, which were in agreement with previous studies [15,24,30,34]. It showed that the pull-out force between post and resin core materials was correlated to the flexural strength (r = 0.398, *p* = 0.0218). In the same way, after pull-out testing, most of the fractures seen on the post surface were mixed and cohesive failures. This pattern also agreed with the findings of previous studies [9,19]. Notably, flexural strength was more positively correlated to root dentin shear bond strength than to the pull-out force between post and resin core materials. Although the root dentin substrate had a uniform composition (though individual differences in human, adhered surface, can be considered), as showed in Table 3. Each post showed a different composition, for example, the difference between materials of quartz and glass fiber, and the difference in this content, difference in matrix composition, and the contraction stress occurring during polymerization, which is considered to be the result. Additionally, the quality of the adhesive interface was affected by the filler content of the core build-up materials, and the self-etching adhesive [9]. By the way, the second test hypothesis was also accepted. Thus, it is likely that multiple linear regression analyses were conducted using the three independent factors of flexural strength, pull-out force and root dentin shear bond strength (*p* = 0.002, Figure 4).

This study further revealed that different variables—such as the type of post, core build-up material, and post surface pretreatment—might affect post-core interfacial strength [6,7,11,15]. Significantly higher forces was recorded for Tokuyama FR Post than another post systems at three times periods. From this study, it is thought that the main reason is that the value of flexural strength of the core material showed the significant highest among those of others. It is also thought that post surface pretreatment had a positive effect on the retentive force of the post (arising from the interaction between the post systems and their pretreatments; Table 1 and Table 2), and thus contributed to their superior performance compared to another post system.

The testing of the immediately after setting condition was to simulate clinical scenarios. However, the clinically commonly used state immediately after curing is extremely fragile as this result shows. Increase in interfacial strength after one-day storage might be caused by the further polymerization of resin core build-up materials. The higher mechanical strength obtained after one-day storage when compared with the immediate condition (except RelyX Unicem 2 Automix) resulted partly from the stiffer and stronger core build-up materials [5,34]. Bonding ability or setting process was enhanced during water storage, and likewise stress relaxation by hygroscopic expansion as a result of water sorption during the storage of resin core build-up materials [17,35,36]. Consequently, hygroscopic expansion resulted in higher interfacial sliding friction, which thus led to higher interfacial strengths of the resin core materials after one-day storage [17].

Apart from the characteristics of restorative materials, the success or failure of core build-up materials is inextricably dependent on the environmental conditions. In the oral cavity, water (as the major component of saliva) plays a major role in filler–matrix bond failures in resin-based materials. It causes filler elements to leach out. It induces filler failures and filler–matrix debonding, thereby compromising the matrix material because debonded fillers may act as stress concentrators and significantly multiply the number of potential crack growth sites [37,38]. It also has a plasticizing effect on the matrix. Apart from water, many failures in practice occur because of thermal stress, especially at the root dentin–core interface and core–pretreated post interface. Due to the dual factors of water immersion and thermal stress, microleakage occurred and lowered interfacial friction, eventually resulting in resin core build-up failure [3,4,5,14,34].

In this study, thermocycling tests of 20,000 thermocycles were carried out because they aptly represent the conditions in the oral cavity [33]. According to the literature [39], provisional estimate of approximately 10,000 cycles per year was suggested. Therefore, we consider the thermocycles 20,000 times to be a two year in vivo condition. After TC 20k, lower values were seen for flexural strength and push-out force when compared with the one-day condition. Hence, half of the resin core build-up systems, including interfacial strengths of the resin core materials, examined in this study could not withstand the damage induced by the TC 20k condition. It is conceivable that the cause of this decreasing in value is as described above.

Self-etch adhesives were used with resin core build-up systems in this study. They function by dissolving or removing the smear layer and by resin tag formation. RelyX Unicem 2 Automix (self-adhesive resin composite cement type) showed the lowest flexural strength and root dentin shear bond strength values among all the studied cements. However, it did not show the lowest pull-out force. The functional phosphate monomers in this cement could be the reason for an effective adhesion to the post (whose main content could be glass fibers; Table 3). The phosphate ester group of the monomer could have bonded directly to the glass fibers, thus creating chemical bonds between the resin composite cement and RelyX FiberPost [15,30,39]. Therefore, it is generally agreed that zirconia restorations should be luted with resin composite cements containing phosphate monomers [40].

To indicate the optimal choice for clinical practice among the plethora of commercially available resin core build-up agents, we propose that their sealing and retentive efficacy be evaluated in order to determine their laboratory and clinical performance.

The limitations of the present study were, we did not actually seal the tooth root with a core material and do a pull-out test with the post, but we would like to do this experiment in the future. It is hoped that this result will support the experimental results of this paper by the relationship between the results of the pull-out test and the shear bending strength of the core material.

## 5. Conclusions

Within the limitation of the study, it can be concluded that the flexural strength of core build-up material is positively correlated with resin-root dentin shear bond strength and pull-out force. Therefore, flexural strength of the core build-up material may be used as the predictor of the shear bond strength resin core build-up to the root dentin and the retentive force of the post.

## Figures and Tables

**Figure 1 polymers-12-02947-f001:**
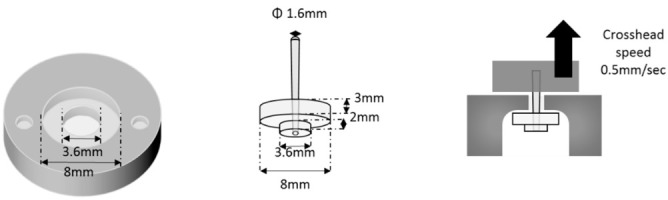
Preparation and testing of pull-out force specimens.

**Figure 2 polymers-12-02947-f002:**
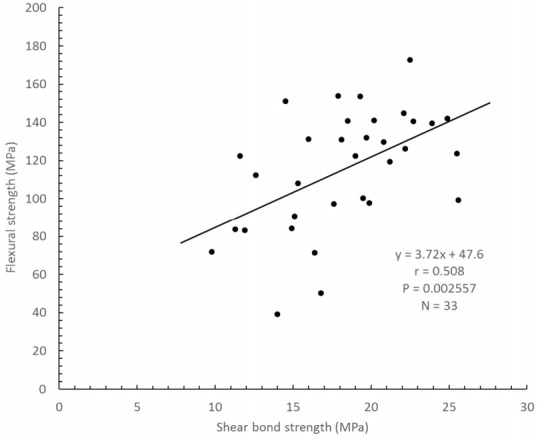
Relationship between flexural strength and root dentin shear bond strength dentin (r = 0.508, *p* = 0.0026, *n* = 33).

**Figure 3 polymers-12-02947-f003:**
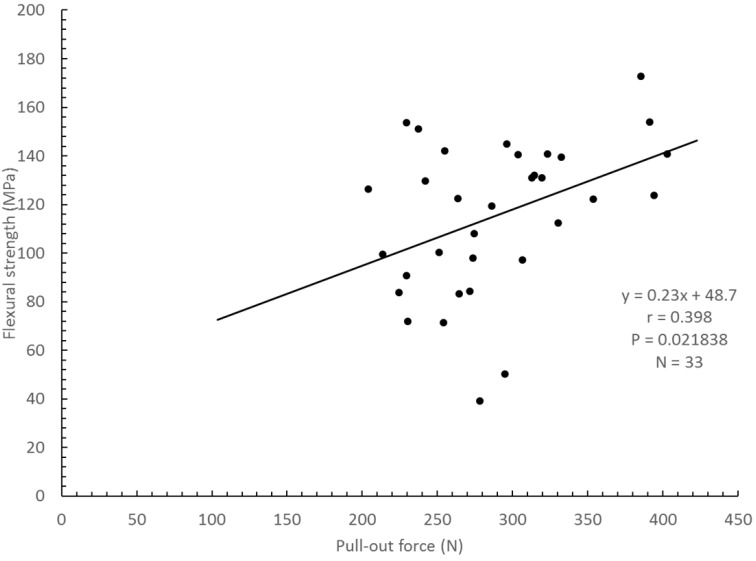
Relationship between flexural strength and pull-out force (r = 0.398, *p* = 0.0218, *n* = 33).

**Figure 4 polymers-12-02947-f004:**
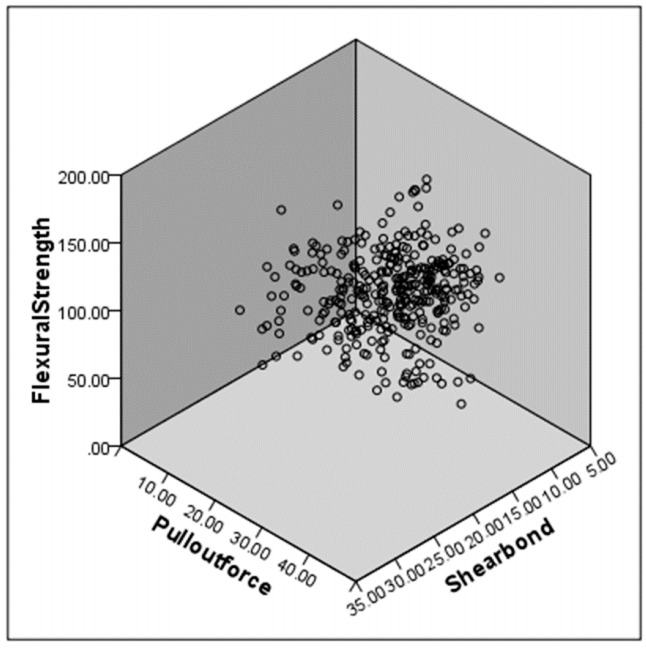
Relationship among flexural strength, pull-out force and root dentin shear bond strength (Flexural strength = 3.264 × Shear bond strength + 1.533 × Pull out bond strength + 10.870, *p* = 0.002, *n* = 33).

**Table 1 polymers-12-02947-t001:** Core materials investigated.

Product	Composition	Manufacturer	Batch No.
FluoroCore 2+	Barium fluoro alumino borosilicate glass (Silanated), Sialne treated silica, Aluminum Oxide, Bis-GMA, Urethane Dimetacrylate, Polymerizable dimetacrylate, Benzoyl Peroxide, Filler content: 69.1 wt.%, 46 vol.%. The particle size ranges from 0.04 to 25 µm.	Dentsply/Caulk, Milford, DE, USA	160415
RelyX Ultimate	Surface treated Glass Powder Filler, Phosphate ester monomer, TEGDMA, 1,12-Dodecane Dimethaycrylate, silica Filler, Initiator, Calcium Hydroxide, Titanium Dioxide, Filler content: About 70 wt.%	3M, Seefeld, Germany	642680
RelyX Unicem 2 Automix	Surface treated Glass Powder Filler, Phosphate ester monomer, TEGDMA, 1,12-Dodecane Dimethaycrylate, Silica Filler, Initiator, Calcium Hydroxide, Sodium *p*-Toluensulfinatet, Methacrylated Amine, Titanium Dioxide, Filler content: About 70 wt.%	3M, Seefeld, Germany	646984
Filtek Bulkfill Flowable Restorative	Silane Treated Ceramic, UDMA, Bis EMA, Bis-GMA, TEGDMA, Other Dimethacrylate, Ytterbium Fluoride, Filler content: 64.5 wt.%, 42.5 vol.%	3M, St. Paul, USA	N815551
NX3	Barium Aluminoborosilicate glass, Ytterbium trifluoride, Fumed Silica, TEGDMA, UDMA, EBPADMA, Initiator, Stabilizer, Filler content: 67.5 wt.%, 43.3 vol.%	Kerr, Orange, CA, USA	6021181
MultiCore Flow	Ytterbium trifluoride, Bis-GMA, UDMA, TEGDMA, Dibenzoyl peroxide, Filler content: 70 wt.%, 46 vol.%, The particle size ranges from 0.04 to 25 µm.	Ivoclar Vivadent AG, Schaan, Liechtenstein	W02582
UniFil Core EM	UDMA, Dimethacrylate, Fluoroaluminosilicate glass, Iron oxide, Dibenzoyl peroxide, Butylated hydroxytoluene, Filler content: 75 wt.%	GC, Tokyo, Japan	1604251
Beauti Core Flow Paste	Glass Powder Filler (S-PRG Filler), Bis-GMA, TEGDMA, Silica, Initiator, Others, Filler content: 60–70 wt.%	Shofu, Kyoro, Japan	61610
i-TFC system Post Resin	Dimethacrylates, Silica, Barium glass filler, Photoinitiators, Stabilizer, Others Filler content: 67 wt.%	Sun Medical, Moriyama, Shiga, Japan	MX13
ESTECORE	Bis-GMA, TEGDMA, Bis-MPEPP, Silica-Zirconia Filler, Camphorquinone, Peroxide, Radial amplifier, Others, Filler content: 75 wt.%	Tokuyama Dental, Tokyo, Japan	112006
Clearfil DC Core Automix ONE	Bis-GMA, TEGDMA, Hydrophilic aliphatic dimethacrylate, Hydrophobic aromatic dimethacrylate, Silanated barium glass filler, Silanated colloidal silica, Colloidal silica, dl-Camphor Quinone, Aluminum oxide filler, Initiators, Accelerators, Pigments. Filler content: 74 wt.%, 52 vol.%	Kuraray Noritake Dental, Tainai, Niigata, Japan	B30218

Bis-GMA: Bisphenol A glycidyl methacrylate, TEGDMA: Tri-ethylene-glycol dimethacrylate, Bis-EMA: Bisphenol A ethoxyl methacrylate, UDMA: Urethane dimethacrylate, EBPADMA: Ethoxylated bis-phenol-A-dimethacrylate, S-PRG: Surface reaction type pre-reacted glass-ionomer filler, Bis-MPEPP: 2,2-Bis(4-methacryloyloxypolyethoxyphenyl) propane.

**Table 2 polymers-12-02947-t002:** Self-etching adhesives and system adhesive components.

Adhesive	Batch No.	Composition	Manufacturer	Surface Treatment of Root Dentin	Surface Treatment of Fiber Post
Prime & Bond XP	14000254	PENTA, TCB resin, UDMA, TEGDMA, HEMA, Nanofiller, Camphorquinone, Butylated benzenediol, Ethyl-4 (dimethylamino) benzoate, Camphorquinone, Functionalised amorphous silica	Dentsply/Caulk Milford, DE, USA	Prime & Bond XP + Self Cure Activator (150306, 20 s) − air (5 s) − FluoroCore 2 ± light (20 s)	To fiber post: Prime & Bond XP + Self Cure Activator − air (5 s)
Scotchbond Universal Adhesive	596935	Bis GMA, HEMA, Decamethylene Dimethacrylate, Silane Treated Silica, Vitrabond Copolymer, MDP, Initiators, Silane, Ethanol	3M, Seefeld, Germany	Scotchbond Universal Adhesive (20 s) − air (5 s) − light (10 s) − RelyX Ultimate − light (20 s). RelyX Unicem 2 Automix: None	Scotchbond Universal Adhesive (20 s) − air (5 s) − light (10 s). RelyX Unicem 2 Automix: None
OptiBond XTR (Primer)	6001208	GPDM, hydrophilic co-monomers, Water, ethanol, acetone, photo-initiator.	Kerr, Orange, CA, USA	OptiBond XTR Primer (20 s) − air (5 s) − OptiBond XTR Adhesive (15 s) − air (5 s) − light (10 s) − NX3 − light (20 s)	Porcelain Primer (091632, Shofu) − air (5 s) − OptiBond XTR Adhesive (15 s) − air (5 s) − light (10 s)
OptiBond XTR (Adhesive)	6001210	HEMA, dimethacrylate monomers, tri-functional methacrylate monomer, ethanol, photo-initiator, bariumaluminosilicate filler, nano-silica, Sodium hexafluorosilicate.	Kerr, Orange, CA, USA		
Adhese Universal	V18786	Methacrylates, Water, Ethanol, Highly dispersed silicon dioxide, Initiators and Stabilisers	Ivoclar Vivadent, Schaan, Liechtenstein	Adhese Universal (20 s) − air (5 s) − light (10 s) − MultiCore Flow − light (20 s)	Monobond Plus (60 s) − air (5 s)
Monobond Plus	U25466	Ethanol, Methacrylated phosphoric acid ester	Ivoclar Vivadent, Schaan, Liechtenstein		
G-Premio Bond	1607212	Acetone, Water, Dimethacrylate, 4-MET, Phosphoric ester monomer, Thiophosphoric ester monomer, Photoinitiator Butylated hydroxytoluene, Silica	GC, Tokyo, Japan	G-Premio BOND + G-Premio Bond DCA (10 s) − air (5 s) − light (10 s) − UniFil Core EM − light (20 s)	Ceramic Primer II– air (5 s)
G-Premio Bond DCA	1603032	Ethanol, Water, Catalyst	GC, Tokyo, Japan		
Beauti Dual Bond EX (A)	41607	Acetone, Water, Initiator, Others	Shofu, Kyoto, Japan	Beauti Dual Bond EX (A + B, 10 s) − air (5 s) − light (10 s) − BeautiCore Flow Paste − light (20 s)	Porcelain Primer (091632, Shofu) − air (5 s)
Beauti Dual Bond EX (B)	11608	Acetone, Bis-GMA, Carboxylic acid monomer, TEGDMA, Initiator, Others	Shofu, Kyoto, Japan		
Porcelain Primer	91632	Ethanol, silane coupling agent, Others	Shofu, Kyoto, Japan		
i-TFC bond	MS 1	Bond: Poly-function (math) metacrylates, Acetone, 4-META, Water, Silica, Others. Bond Brush: Aromatic amine, Sodium *p*-toluenesulfinate.	Sun Medical, Moriyama, Japan	i-TFC bond + i-TFC bond Brush (20 s) − air (5–10 s) − i-TFC system Post Resin − light (20 s)	i-TFC system FiberPost Primer (MF 1)
BONDMER lightless	7067	Liquid A: Phosphoric acid monomer (New 3D-SR monomer), MTU 6, HEMA, Bis-GMA, TEGDMA, Acetone, Others. Liquid B: γ-MPTES, Borate, Peroxide, Acetone, Isopropyl alcohol, Water, Others	Tokuyama Dental, Tokyo, Japan	BONDMER lightless (1–2 s) − air (5 s)	BONDMER lightless (1–2 s) − air (5 s)
Clearfil Universal Bond Quick	B30218	10-Methacryloyloxydecyl dihydrogen phosphate (10-MDP), HEMA, Bis-GMA, Ethanol, Water, Hydrophilic amide monomers, Colloidal silica, Sodium fluoride, dl-Camphorquinone.	Kuraray Noritake Dental, Tainai, Niigata, Japan	Clearfil Universal Bond Quick (1–2 s) − air (5 s) − light (10 s) − Clearfil DC Core Automix ONE − light (20 s)	Clearfil Universal Bond Quick + Clearfil Porcelain Bond Activator − air (5 s)
Clearfil Porcelain Bond Activator	9D0033	3-MPTS, Hydrophobic aromatic dimethacrylate	Kuraray Noritake Dental, Tainai, Niigata, Japan		

PENTA: Phosphoric acid modified acrylate resin, TCB resin: Carboxylic acid modified dimethacrylate, UDMA: Urethane dimethacrylate, TEGDMA: Triethyleneglycol dimethacrylate, HEMA: 2-Hydroxyethylmethacrylate, Bis-GMA: Bisphenol A glycidyl methacrylate, 10-MDP: 10-Methacryloyloxydecyl dihydrogen phosphate, GPDM: Glycerophosphatedimethacrylate, 4-MET: 4-methacryloxyethyl trimellitic acid, 4-META: 4-methacryloxyethyl trimellitate anhydrine, MTU-6: 6-methacryloxyhexyl 2-thiouracil-5-carboxylate, γ-MPTES: 3-(triethoxysilyl) propyl methacrylate, 3-MPTS: 3-Methacryloxypropyltrimethoxysilane.

**Table 3 polymers-12-02947-t003:** Fiber posts investigated.

Product (Diameter)	Composition	Manufacturer	Batch No.
FluoroPost (1.6 mm)	Quartz fibers 60 vol.%, Epoxy resin vol.40%	Dentsply/Caulk, Milford, DE, USA	160613
RelyX FiberPost (1.6 mm)	Glass fibers, Composite resin matrix	3M, Seefeld, Germany	306831603
FRC Postec Plus (1.6 mm)	Glass fibers, Dimethacrylates, Ytterbium fluoride	Ivoclar Vivadent AG, Schaan, Liechtenstein	V30339
GC Fiber Post (1.6 mm)	Glass fibers, Metacrylate	GC, Tokyo, Japan	1609021
BeautiCore FiberPost (1.6 mm)	Glass fiber, Copolymer of Bis-GMA and Methacrylic ester monomer	Shofu, Kyoro, Japan	41601
i-TFC system Fiber Post (1.6 mm)	Glass fiber, Optical fiber, others	Sun Medical, Moriyama, Shiga, Japan	ML2
Tokuyama FR Post (1.6 mm)	Glass fibers, Colpolymer Bis-GMA resin	Tokuyama Dental, Tokyo, Japan	1604251
Clearfil AD Fiber Post (1.6 mm)	Glass fibers, Colpolymer Bis-GMA and Methacrylic acid monomer	Kuraray Noritake Dental, Tainai, Niigata, Japan	7U0001

**Table 4 polymers-12-02947-t004:** Flexural strength of various Core build-up materials and luting materials (MPa, mean (S.D.)).

		Time	
	Immediate	After One-Day Storage	TC 20k
FluoriCore 2	83.3 (8.8)	132.0 (8.4)	122.4 (10.6)
RelyX Ultimate	71.4 (4.6)	119.4 (3.6)	100.3 (4.9)
RelyX Unicem 2 Automix	71.9 (5.7)	108.0 (6.8)	83.8 (5.1)
Filtek BulkFill Flowable Restorative	50.3 (1.8)	144.9 (5.3)	129.7 (8.3)
NX3	39.1 (5.2)	123.7 (9.8)	97.9 (8.7)
MultiCore Flow	99.4 (7.4)	142.1 (9.1)	126.3 (8.2)
UniFil Core EM	90.8 (7.3)	153.6 (11.4)	151.2 (12.1)
BeautiCore Flow Paste	112.4 (9.3)	140.7 (7.9)	131.1 (5.8)
i-TFC system Post Resin	84.3 (4.1)	139.4 (6.4)	131.0 (7.6)
ESTECORE	122.3 (9.1)	172.8 (10.2)	153.9 (13.3)
Clearfil DC Core Automix ONE	97.3 (19.4)	140.6 (9.6)	140.9 (8.6)

TC 20k: After 20,000 thermocycles.

**Table 5 polymers-12-02947-t005:** Comparison of means (Tukey HSD Procedure) for flexural strength of each material with regard to time (superscript letters represent groups not significantly different, *p* > 0.05).

FluoroCore 2	RelyX Ultimate	RelyX Unicem 2 Automix	Filtek BulkFill Flowable Restorative	NX3	MultiCore Flow	UniFil Core EM	BeautiCore Flow Paste	i-TFC System Post Resin	ESTECORE	Clearfil DC Core Automix One
Immediate	Immediate	Immediate ^b^	Immediate	Immediate	Immediate	Immediate	Immediate	Immediate	Immediate	Immediate
TC 20k ^a^	TC 20k	TC 20k ^b^	TC 20k	TC 20k	TC 20k	TC 20k ^c^	TC 20k ^d^	TC 20k ^e^	TC 20k	One-day ^f^
One-day ^a^	One-day	One-day	One-day	One-day	One-day	One-day ^c^	One-day ^d^	One-day ^e^	One-day	TC 20k ^f^

TC 20k: After 20,000 thermocycles.

**Table 6 polymers-12-02947-t006:** Comparison of means (Tukey HSD procedure) for flexural strength of material at each time (superscript letters represent groups not significantly different, *p* > 0.05).

Immediately	After One-Day Storage	TC 20k
NX3 ^a^	RelyX Unicem 2 Automix ^j^	RelyX Unicem 2 Automix
Filtek BulkFill Flowable Restorative ^a^	RelyX Ultimate ^j k^	NX3 ^o^
RelyX Ultimate ^b^	NX3 ^k^	RelyX Ultimate ^o^
RelyX Unicem 2 Automix ^b^	FluoriCore 2 ^k l^	FluoriCore 2 ^p^
FluoriCore 2 ^b c^	i-TFC system Post Resin ^l m^	MultiCore Flow ^p^
i-TFC system Post Resin ^b c d e^	Clearfil DC Core Automix ONE ^l m n^	Filtek BulkFill Flowable Restorative ^p q^
UniFil Core EM c d e f	BeautiCore Flow Paste ^l m n^	i-TFC system Post Resin ^p q^
Clearfil DC Core Automix ONE ^d e f g^	MultiCore Flow ^l m n^	BeautiCore Flow Paste ^p q^
MultiCore Flow ^f g h^	Filtek BulkFill Flowable Restorative ^l m n^	Clearfil DC Core Automix ONE ^q r^
BeautiCore Flow Paste ^h i^	UniFil Core EM ^n^	UniFil Core EM ^r^
ESTECORE ^i^	ESTECORE	ESTECORE ^r^

TC 20k: After 20,000 thermocycles.

**Table 7 polymers-12-02947-t007:** Shear bond strength to root dentin of various resin core build-up and luting materials (MPa, mean (S.D.), Adh.).

		Time	
	Immediately	After One-Day Storage	TC 20k
FluoriCore 2	11.9 (3.6, 0)	19.7 (2.3, 0)	11.6 (2.9, 0)
RelyX Ultimate	16.4 (3.8, 0)	21.2 (4.9, 0)	19.5 (3.3, 0)
RelyX Unicem 2 Automix	9.8 (2.6, 0)	15.3 (5.3, 0)	11.3 (2.3, 0)
Filtek BulkFill Flowable Restorative	16.8 (2.5, 0)	22.1 (3.3, 0)	20.8 (3.2, 0)
NX3	14.0 (2.1, 0)	25.5 (3.5, 0)	19.9 (3.6, 0)
MultiCore Flow	25.6 (4.2, 0)	24.9 (4.3, 0)	22.2 (4.4, 0)
UniFil Core EM	15.1 (3.2, 0)	19.3 (5.3, 0)	14.5 (3.2, 0)
BeautiCore Flow Paste	12.6 (2.4, 0)	18.5 (3.9, 0)	16.0 (3.2, 0)
i-TFC system Post Resin	14.9 (2.4, 0)	23.9 (4.1, 0)	18.1 (4.0, 0)
ESTECORE	19.0 (4.9, 0)	22.5 (4.3, 0)	17.9 (3.2, 0)
Clearfil DC Core Automix ONE	17.6 (4.0, 0)	22.7 (4.0, 0)	20.2 (4.0, 0)

*n* = 10, Adh: Number of adhesive failure modes, TC 20k: After 20,000 thermocycles.

**Table 8 polymers-12-02947-t008:** Comparison of means (Tukey HSD Procedure) for shear bond strength to root dentin of each material with regard to time (superscript letters represent groups not significantly different, *p* > 0.05).

FluoroCore 2	RelyX Ultimate	RelyX Unicem 2 Automix	Filtek BulkFill Flowable Restorative	NX3	MultiCore Flow	UniFil Core EM	BeautiCore Flow Paste	i-TFC System Post Resin	ESTECORE	Clearfil DC Core Automix ONE
TC 20k ^a^	Immediate ^b^	Immediate ^c^	Immediate ^d^	Immediate	TC 20k ^f^	TC 20k ^g^	Immediate ^h^	Immediate ^i^	TC 20k ^j^	Immediate ^k^
Immediate ^a^	TC 20k ^b^	TC 20k ^c^	TC 20k ^d^	TC 20k ^e^	One-day ^f^	Immediate ^g^	TC 20k ^h^	TC 20k ^i^	Immediate ^j^	TC 20k ^k^
One-day	One-day ^b^	One-day ^c^	One-day ^d^	One-day ^e^	Immediate ^f^	One-day ^g^	One-day ^h^	One-day	One-day ^j^	One-day ^k^

TC 20k: After 20,000 thermocycles.

**Table 9 polymers-12-02947-t009:** Comparison of means (Tukey HSD procedure) for shear bond strength to root dentin of material at each time (superscript letters represent groups not significantly different, *p* > 0.05).

Immediately	After One-Day Storage	TC 20k
RelyX Unicem 2 Automix ^a^	RelyX Unicem 2 Automix ^d^	RelyX Unicem 2 Automix ^g^
FluoriCore 2 ^a b^	BeautiCore Flow Paste ^d e^	FluoriCore 2 ^g h^
BeautiCore Flow Paste ^a b^	UniFil Core EM ^d e f^	UniFil Core EM ^g h i^
NX3 ^a b c^	FluoriCore 2 d ^e f^	BeautiCore Flow Paste ^g h i j^
i-TFC system Post Resin ^a b c^	RelyX Ultimate ^d e f^	ESTECORE ^h i j^
UniFil Core EM ^a b c^	Filtek BulkFill Flowable Restorative ^e f^	i-TFC system Post Resin ^i j^
RelyX Ultimate ^b c^	ESTECORE ^e f^	RelyX Ultimate ^i j^
Filtek BulkFill Flowable Restorative ^b c^	Clearfil DC Core Automix ONE ^e f^	NX3 ^i j^
Clearfil DC Core Automix ONE ^b c^	i-TFC system Post Resin ^e f^	Clearfil DC Core Automix ONE ^i j^
ESTECORE ^c^	MultiCore Flow ^f^	Filtek BulkFill Flowable Restorative ^i j^
MultiCore Flow	NX3 ^f^	MultiCore Flow ^j^

TC 20k: After 20,000 thermocycles.

**Table 10 polymers-12-02947-t010:** Pull-out force between various resin-core build-up materials or luting cement and FiberPost (N, mean (S.D.), Adh.).

Materials/Fiber Post (Pretreating Agent)		Time	
	Immediately	After One-Day Storage	TC 20k
FluoriCore 2/Fluoropost (XP Bond + Self cure Activator)	264.6 (40.2, 1)	314.6 (40.2, 2)	263.6 (33.3, 0)
RelyX Ultimate/RelyX Fiber Post (Scotchbond Universal Adhesive)	253.8 (40.2, 2)	286.2 (25.5, 2)	250.9 (26.5, 1)
RelyX Unicem 2 Automix/RelyX Fiber Post (None)	230.3 (22.5, 1)	274.4 (14.7, 0)	224.4 (30.1, 1)
Filtek BulkFill Flowable Restorative/RelyX Fiber Post (Scotchbond Universal Adhesive)	295.0 (61.7, 1)	296.0 (31.4, 0)	242.1 (28.4, 0)
NX3/BeautiCore Fiber (Porcelain Primer (Shofu) + OptiBond XTR (Adhesive))	278.3 (47.0, 1)	394.0 (44.1, 2)	273.4 (45.1, 2)
MultiCore Flow/FRC Postec Plus (Monobond Plus)	213.6 (34.3, 0)	254.8 (19.6, 0)	203.8 (34.3, 0)
UniFil Core EM/GC Fiber Post (Ceramic Primer II)	229.3 (40.2, 0)	229.3 (28.4, 0)	237.2 (26.5, 0)
BeautiCore Flow Paste/BeautiCore FiberPost (Shofu Porcelain Primer)	330.3 (46.1, 1)	402.8 (31.4, 1)	312.6 (75.5, 0)
i-TFC system Post Resin/i-TFC system Fiber Post (i-TFC system Fiber Post Primer)	271.5 (41.2, 1)	332.2 (39.2, 0)	319.5 (49.0, 0)
ESTECORE/Tokuyama FR Post (BONDMER Lightless)	353.8 (32.3, 0)	385.1 (34.3, 0)	391.0 (25.5, 0)
Clearfil DC Core Automix ONE/Clearfil AD Fiber Post (Universal Bond Quick + Porcelain Activator)	306.7 (38.2, 1)	303.8 (29.4, 1)	323.4 (41.2, 0)

*n* = 10, Adh: Number of adhesive failure modes, TC 20k: After 20,000 thermocycles.

**Table 11 polymers-12-02947-t011:** Comparison of means (Tukey HSD Procedure) for pull-out force of each material with regard to time (superscript letters represent groups not significantly different, *p* > 0.05).

FluoroCore 2	RelyX Ultimate	RelyX Unicem 2 Automix	Filtek BulkFill Flowable Restorative	NX3	MultiCore Flow	UniFil Core EM	BeautiCore Flow Paste	i-TFC System Post Resin	ESTECORE	Clearfil DC Core Automix ONE
TC 20k ^a^	TC 20k ^b^	TC 20k ^c^	TC 20k	TC 20k ^e^	TC 20k ^f^	Immediate ^g^	TC 20k ^h^	Immediate ^i^	Immediate ^j^	One-day ^k^
Immediate ^a^	Immediate ^b^	Immediate ^c^	Immediate ^d^	Immediate ^e^	Immediate ^f^	One-day ^g^	Immediate ^h^	TC 20k ^i^	One-day ^j^	Immediate ^k^
One-day	One-day ^b^	One-day ^c^	One-day ^d^	One-day ^e^	One-day ^f^	TC 20k ^g^	One-day	One-day ^i^	TC 20k ^j^	TC 20k ^k^

TC 20k: After 20,000 thermocycles.

**Table 12 polymers-12-02947-t012:** Comparison of means (Tukey HSD procedure) for pull-out force of material (core or cement materials/fiber post) at each time (superscript letters represent groups not significantly different, *p* > 0.05).

Immediately	After One-Day Storage	TC 20k
MultiCore Flow/FRC Postec Plus ^a b^	UniFil Core EM/GC Fiber Post ^f^	MultiCore Flow/FRC Postec Plus
UniFil Core EM/GC Fiber Post ^a b^	MultiCore Flow/FRC Postec Plus ^f g h^	RelyX Unicem 2 Automix/RelyX Fiber Post ^l^
RelyX Unicem 2 Automix/RelyX Fiber Post ^a b^	RelyX Unicem 2 Automix/RelyX Fiber Post ^f g h^	UniFil Core EM/GC Fiber Post ^l m^
RelyX Ultimate/RelyX Fiber Post ^a b c^	RelyX Ultimate/RelyX Fiber Post ^f g^	Filtek BulkFill Flowable Restorative/RelyX Fiber Post ^l m^
FluoriCore 2/Fluoropost ^a b c d^	Filtek BulkFill Flowable Restorative/RelyX Fiber Post ^f g h i^	RelyX Ultimate/RelyX Fiber Post ^l m n^
i-TFC system Post esin/i-TFC system Fiber Post ^a b c d^	Clearfil DC Core Automix ONE/Clearfil AD Fiber Post ^g h i^	FluoriCore 2/Fluoropost ^l m n o^
NX3/BeautiCore Fiber ^a b c d^	FluoriCore 2/Fluoropost ^g h i^	NX3/BeautiCore Fiber ^l m n o^
Filtek BulkFill Flowable Restorative/RelyX Fiber Post ^b c d e^	i-TFC system Post Resin/i-TFC system Fiber Post ^h i j^	BeautiCore Flow Paste/BeautiCore FiberPost ^n o^
Clearfil DC Core Automix ONE/Clearfil AD Fiber Post ^c d e^	ESTECORE/Tokuyama FR Post ^j k^	i-TFC system Post Resin/i-TFC system Fiber Post ^n o^
BeautiCore Flow Paste/BeautiCore FiberPost ^d e^	NX3/BeautiCore Fiber ^j k^	Clearfil DC Core Automix ONE/Clearfil AD Fiber Post ^o p^
ESTECORE/Tokuyama FR Post ^e^	BeautiCore Flow Paste/BeautiCore FiberPost ^k^	ESTECORE/Tokuyama FR Post ^p^

TC 20k: After 20,000 thermocycles.

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
