# Peer review of "Flexural Strength of Resin Core Build-Up Materials: Correlation to Root Dentin Shear Bond Strength and Pull-Out Force"

_polymers, 2020, doi:10.3390/polym12122947_

Round 1
Reviewer 1 Report
I have reviewed the manuscript “Flexural strength of resin core build-up materials: correlation to root dentin bond strength and pull-out force” submitted to “Polymers” for publication. I found this work interesting and fit well with in the scope of this journal. In this study, authors have investigated the effects of root dentin bond strength and pull-out force of resin core build-up materials on flexural strength at three time periods. This is a well-designed and well conducted study and the manuscript fits well within the scope of the journal; it needs some major improvements; there are a few suggestions that authors may consider to improve it further:
The use of English language is reasonable, however, there are a number of punctuation and grammatical errors; that should be corrected and rephrased using academic English for a better flow of text for reader.
Abstract: Line 23: “aims of this investigation were to investigate the….” Please rephrase to replace the repetitive word.
The conclusive statement should be added to the abstract
There are several spaces (double spaces in the text) that should be removed. Use one style for p-value: p or P (such as line 37)?
Line 83-84 needs to be rephrased.
For premolars extracted used is the study; was there any exclusion criteria; such as teeth with caries facture line, worn etc were excluded or not?
Results and data have been presented very comprehensively for which I must appreciate authors’ efforts and hard work
Discussion lines 353-358: the information presented should have reference citation; author may consider the following recent studies to support this information
Influence of various specimen storage strategies on dental resin-based composite properties." Materials Technology (2020): 1-9.
Effects of deformation rate variation on biaxial flexural properties of dental resin composites." Journal of Taibah University medical sciences 13.4 (2018): 319-326.
What are the limitation of this study? Should be included.
The conclusive remarks should be described more precisely.
Author Response
This section should more be focused: please describe DBS, post, resin composites used in post cementation and built-up. This should give a clear background introducing the aim of the study. You might use i.e:The use of English language is reasonable, however, there are a number of punctuation and grammatical errors; that should be corrected and rephrased using academic English for a better flow of text for reader.
Masao: Thank you for your suggestion. Please see the revised manuscript.
Abstract: Line 23: “aims of this investigation were to investigate the….” Please rephrase to replace the repetitive word.
Masao: The correction text is shown on Line 23-25. It was displayed with a yellow marker.
The conclusive statement should be added to the abstract
Masao: The correction text is shown on Line 39-42. It was displayed with a yellow marker.
There are several spaces (double spaces in the text) that should be removed. Use one style for p-value: p or P (such as line 37)?
Masao: I am very sorry. You are right. Please show the revised manuscript. Especially, Line 37, as displayed with a yellow marker.
Line 83-84 needs to be rephrased.
Masao: The correction text is shown on Line 82-83. It was displayed with a yellow marker. Thanks.
For premolars extracted used is the study; was there any exclusion criteria; such as teeth with caries facture line, worn etc were excluded or not?
Masao: YES. Naturally, worn etc were excluded. The correction text is shown on Line 115-116. It was displayed with a yellow marker. Thanks.
Results and data have been presented very comprehensively for which I must appreciate authors’ efforts and hard work.
Masao: I understand you. However,
- For example; In box and whisker plot graphs, I thought it would be difficult to compare at a glance from both times and products, so I expressed it as a table that I thought it was possible at a glance. So I have tabulated a detailed statistical comparison of times and products. Like the reference [8].
- In box and whisker plot graphs. Individual detailed numbers are difficult to understand, so they are represented as a table to help expert readers understand this difference in detail.
- For example, I would like to see this reference [32].
Therefore, I would like to understand the above reasons.
Discussion lines 353-358: the information presented should have reference citation; author may consider the following recent studies to support this information
Influence of various specimen storage strategies on dental resin-based composite properties." Materials Technology (2020): 1-9.
Effects of deformation rate variation on biaxial flexural properties of dental resin composites." Journal of Taibah University medical sciences 13.4 (2018): 319-326.
Masao: Many thanks for the useful suggestion. It was added as references 37, 38 as you pointed out. Please show on Line 361. Please see it because it is indicated by a yellow marker.
What are the limitation of this study? Should be included.
Masao: Many thanks for the useful suggestion. Please show on Line 386-389. Please see it because it is indicated by a yellow marker. Thanks
The conclusive remarks should be described more precisely.
Masao: The correction text is shown on Line 392-396. It is indicated by a yellow marker. Thanks.
Finally, thank you for your helpful advice. And then I hope you agree with my response.
Reviewer 2 Report
The manuscript presents an interesting issue. However, it is written in a bit chaotic manner. Moreover, English demands editing.
Introduction
This section should more be focused: please describe DBS, post, resin composites used in post cementation and built-up. This should give a clear background introducing the aim of the study. You might use i.e:
Mishra L. et al. Effects of Surface Treatments of Glass Fiber-Reinforced Post on Bond Strength to Root Dentine: A Systematic Review. MATERIALS; 13; 8: 2020
Alshali, Ruwaida Z. et al. Long-term sorption and solubility of bulk-fill and conventional resin-composites in water and artificial saliva. JOURNAL OF DENTISTRY; 43 ; 12 ; 2015
Sokolowski K. et al. Contraction and Hydroscopic Expansion Stress of Dental Ion-Releasing Polymeric Materials. POLYMERS; 10;10: 2018
Al Sunbul, Hanan et al. Polymerization shrinkage kinetics and shrinkage-stress in dental resin-composites .DENTAL MATERIALS ; . 32 ; 8 ; 2016
Line 62 ‘Fiber posts are deep and narrow…’ – I do not understand the sentence
Lines 71-76 do no add any value to the introduction.
M&M
Results
Please present table 4 and 7 and 10 as box and whisker plot graphs.
Tables 5, 6, 8,9, 11, 12 are hard to read, please present data in more clear way or describe it.
Discussion
Please explain why flexural strength was positively correlated to pull-out test and SBS. What phenomenon is behind it?
Why most of the fracture modes seen after SBS were mixed and cohesive failures? How it can be explained.
Please enrich this section in result interpretation and data from other more recent articles i.e.:
Mishra L. et al. Effects of Surface Treatments of Glass Fiber-Reinforced Post on Bond Strength to Root Dentine: A Systematic Review. MATERIALS; 13; 8: 2020
Alshali, Ruwaida Z. et al. Long-term sorption and solubility of bulk-fill and conventional resin-composites in water and artificial saliva. JOURNAL OF DENTISTRY; 43 ; 12 ; 2015
Sokolowski K. et al. Contraction and Hydroscopic Expansion Stress of Dental Ion-Releasing Polymeric Materials. POLYMERS; 10;10: 2018
Al Sunbul, Hanan et al. Polymerization shrinkage kinetics and shrinkage-stress in dental resin-composites .DENTAL MATERIALS ; . 32 ; 8 ; 2016
Conclusion
This section should be added.
Author Response
Introduction
This section should more be focused: please describe DBS, post, resin composites used in post cementation and built-up. This should give a clear background introducing the aim of the study. You might use i.e:
Mishra L. et al. Effects of Surface Treatments of Glass Fiber-Reinforced Post on Bond Strength to Root Dentine: A Systematic Review. MATERIALS; 13; 8: 2020
Alshali, Ruwaida Z. et al. Long-term sorption and solubility of bulk-fill and conventional resin-composites in water and artificial saliva. JOURNAL OF DENTISTRY; 43 ; 12 ; 2015
Sokolowski K. et al. Contraction and Hydroscopic Expansion Stress of Dental Ion-Releasing Polymeric Materials. POLYMERS; 10;10: 2018
Al Sunbul, Hanan et al. Polymerization shrinkage kinetics and shrinkage-stress in dental resin-composites .DENTAL MATERIALS ; . 32 ; 8 ; 2016
Masao: Many thanks for the useful suggestion. It was added as references 3-5, 34 you pointed out. Please show on Line 50, 52, 83, 351. Please see it because it is indicated by a yellow marker.
Line 62 ‘Fiber posts are deep and narrow…’ – I do not understand the sentence
Masao: Since it was described in revised Reference No. 20-22, it was cited. Considering the situation of fiver posts, this meaning is convincing.
Lines 71-76 do no add any value to the introduction.
Masao: We have come up with this idea from previous reports and support it, so we have included it in the introduction.
M&M
Results
Please present table 4 and 7 and 10 as box and whisker plot graphs.
Tables 5, 6, 8,9, 11, 12 are hard to read, please present data in more clear way or describe it.
Masao:
- The Tables in the manuscript for revision are too small to read. That would have made it difficult to read.
- In box and whisker plot graphs, I thought it would be difficult to compare at a glance from both times and products, so I expressed it as a table that I thought it was possible at a glance. So I have tabulated a detailed statistical comparison of times and products. Like the reference [8].
- In box and whisker plot graphs. Individual detailed numbers are difficult to understand, so they are represented as a table to help expert readers understand this difference in detail.
- For example, I would like to see this reference [32].
- Maybe, European likes box and whisker plot graphs.
Discussion
Please explain why flexural strength was positively correlated to pull-out test and SBS. What phenomenon is behind it?
Masao: This idea is the original of this manuscript. It is considered that the mechanical retention force has a great influence on the pull-out test and SBS, and as a result, the flexural strength is evaluated from our report as a method for evaluating the mechanical retention force (our references: 22, 27, 29, 30, 32). So we discussed the relationship between the two results.
Why most of the fracture modes seen after SBS were mixed and cohesive failures? How it can be explained.
Masao: I am trying to measure the bond strength at the interface between root-dentin and core material, but it does not rupture easily at the interface. As you know, it breaks at the weakest point, so it is a break of mix failure and cohesive failure. That is, the mechanical strength of core material selves is weaker than the bonding ability to root-dentin substrate. This is shown in our previous report (our references: 25, 30, 32, 33, 36).
Please enrich this section in result interpretation and data from other more recent articles i.e.:
Mishra L. et al. Effects of Surface Treatments of Glass Fiber-Reinforced Post on Bond Strength to Root Dentine: A Systematic Review. MATERIALS; 13; 8: 2020
Alshali, Ruwaida Z. et al. Long-term sorption and solubility of bulk-fill and conventional resin-composites in water and artificial saliva. JOURNAL OF DENTISTRY; 43 ; 12 ; 2015
Sokolowski K. et al. Contraction and Hydroscopic Expansion Stress of Dental Ion-Releasing Polymeric Materials. POLYMERS; 10;10: 2018
Al Sunbul, Hanan et al. Polymerization shrinkage kinetics and shrinkage-stress in dental resin-composites .DENTAL MATERIALS ; . 32 ; 8 ; 2016
Masao: Many thanks for the useful suggestion. It was added as references 3-5, 34 you pointed out. Please show on Line 50, 52, 83, 351, 365. Please see it because it is indicated by a yellow marker.
Conclusion
This section should be added.
Masao: Many thanks for the useful suggestion. The correction text is shown on Line 391-396. It is indicated by a yellow marker. Thanks.
Finally, thank you for your helpful advice. And then I hope you agree with my response.
Round 2
Reviewer 1 Report
Many thanks for the revision and incorporating all suggested changes to the manuscript
Author Response
Please show the files: polymers-1010127_reply to review 1-2, and polymers-1010127_manuscript for revision -2.
Reviewer 2 Report
The text has been improved. However, I have some minor suggestions:
Newly added text demands English editing.
Please change in the whole text – bond strength to shear bond strength
Aim
Please correct the aim (and add it to the main text), i.e.:
The aim was to investigate correlation between shear bond strength, pull-out force of resin core build-up to root dentin and flexural strength.
Discussion
Lines 386-389 - Please change to:
The limitations of the present study were:…..
Conclusions
Please correct (and add it to the main text), i.e.:
Within the limitation of the study, it can be concluded that the flexural strength of core build-up material is positively correlated with resin-root dentin shear bond strength and pull-out force. Therefore, flexural strength of the core build-up material may be used as the predictor of the shear bond strength resin core build-up to the root dentin and the retentive force of the post.
Author Response
Please show the two files: polymers-1010127_reply to review 2-2, and polymers-1010127_manuscript for revision-2.
